# TextGaze: Gaze-Controllable Face Generation with Natural Language

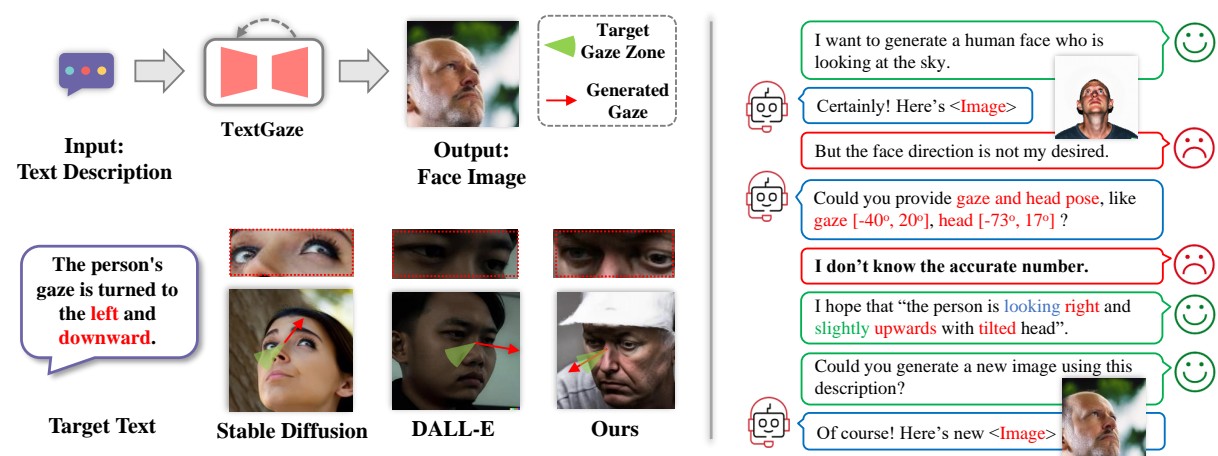

**Figure 1: Left: Overview of our method. Generating gaze images with natural language. Our method significantly outperforms Stable Diffusion and DALL-E. Right: Our motivation. People tend to describe head pose and gaze direction using direction and extent words instead of labels.**

## ABSTRACT

Generating face image with specific gaze information has attracted considerable attention. Existing approaches typically input gaze values directly for face generation, which is unnatural and requires annotated gaze datasets for training, thereby limiting its application. In this paper, we present a novel gaze-controllable face generation task. Our approach inputs textual descriptions that describe human gaze and head behavior and generates corresponding face images. Our work first introduces a text-of-gaze dataset containing over 90k text descriptions spanning a dense distribution of gaze and head poses. We further propose a gaze-controllable text-to-face method. Our method contains a sketch-conditioned face diffusion module and a model-based sketch diffusion module. We define a face sketch based on facial landmarks and eye segmentation map. The face diffusion module generates face images from the face sketch, and the sketch diffusion module employs a 3D face model to generate face sketch from text description. Experiments on the FFHQ dataset show the effectiveness of our method. We will release our dataset and code for future research.

## CCS CONCEPTS

• **Computing methodologies → Computer vision tasks**.

## KEYWORDS

Text-to-Image Generation, Diffusion model, Gaze controlling

## 1 INTRODUCTION

Face image generation has made significant progress in recent years, owing to the remarkable capabilities of adversarial networks [7] and diffusion models [25, 27]. It has various applications including virtual reality [42, 43], digital human [13, 35] and CG film-making [1, 37]. The text-to-face task, which aims to control the face generation using natural language descriptions, has also gained considerable attention [12, 33]. This task generates face images corresponding to input text, attracting interest due to its natural interaction and vast practical potential.

Recent text-to-face methods typically utilize natural language to manipulate semantic feature generation, such as "black hair" and "young boy". These semantic features are manually annotated for each images and provides paired data for the generation model training. In addition to these semantic features, human behaviors are also crucial components evident in human faces. Human gaze reflects human intention and is highly demanded in numerous applications. However, to the best of our knowledge, there is no work for the text-driven gaze-controllable face generation.

Gaze-controllable face generation methods aim to produce face images that correspond to a specified gaze input. Recent approaches typically involve inputting a specific numerical gaze value to generate the corresponding face images [10, 29]. However, we argue that

**Table 1: We list some text-image datasets in various tasks. Our work provides the first text-gaze dataset ToG, containing large and diverse text descriptions.**

| Datasets | Task | #Texts |
|---|---|---|
| BABEL [22] | Text-Body | 63,000 |
| HumanML3D [8] | Text-Body | 44,970 |
| PoseScript [4] | Text-Body | 300,000 |
| Text2FaceGAN [20] | Text-Face | 60,000 |
| CelebAText-HQ [34] | Text-Face | 150,100 |
| **ToG (Ours)** | **Text-Gaze** | **95,548** |

using gaze values as input may not be user-friendly, particularly for non-expert users as shown in Fig. 1. Representing gaze through numerical values could pose challenges for individuals unfamiliar with the technical aspects of gaze estimation. This raises the need for a more user-friendly and intuitive approach to specifying gaze in the generation process.

In this work, we propose a novel gaze-controllable text-to-face generation task, where the input is the text descriptions of human gaze behavior, such as "the person looks forward". Unlike using gaze values, employing natural language is more aligned with common expressive habits, making it more accessible and acceptable to a broader range of users. However, a key challenge arises due to the scarcity of text descriptions of gaze behavior. The development of an accurate and diverse text description dataset is highly demanded.

Our approach decomposes human gaze into eye and head rotations, employing uniform sampling to obtain a dense gaze distribution. Annotating text descriptions for each gaze sample manually is time-intensive, and the expertise of annotators greatly influences the annotation outcomes. Inspired by the success of Large Language Models (LLMs) in natural language expression, we utilize LLMs to annotate gaze behaviors. Prompt construction based on gaze samples and LLM utilization enable the generation of diverse text descriptions.

We first introduce a text of gaze (ToG) dataset containing over 90$k$ gaze descriptions. Our approach decomposes human gaze into eye and head rotations, employing uniform sampling to obtain a dense gaze distribution. Annotating text descriptions for each gaze sample manually is time-consuming, and the expertise of annotators greatly influences the annotation outcomes. Inspired by the success of Large Language Models (LLMs) in natural language, we utilize LLMs to annotate gaze behaviors. We construct prompts based on gaze samples and employ LLM to generate diverse text descriptions.

We further introduce a gaze-controllable text-to-face generation method, consisting of a sketch-conditioned face diffusion module and a model-based sketch diffusion model. Our approach utilizes CLIP to obtain feature embeddings from text and introduces a text attention module to extract gaze-related and head-related features from these embeddings. By learning gaze patterns from the extracted features, we convert gaze patterns into face sketches based on a 3D face model. The face diffusion model then generates face images from these sketches. Unlike conventional methods that rely on annotated gaze datasets for training, our method generates diverse face images without the need for such datasets, using face sketches instead.

Overall, our contributions are three-fold.

- We introduce a novel gaze controllable face generation task where the input is a text description of human gaze behavior. We provide the first text-to-gaze dataset containing more than 90k descriptions. LLMs are used to produce accurate and diverse annotations. To the best of our knowledge, our work is the first to leverage LLMs for gaze image generation.
- We propose a two-stage text-to-face method. Our method leverages the prior information from 3D face models, eliminating the need for gaze dataset. We also propose a text attention module. The module aggregates gaze and head information from text feature embedding, and improve the performance of gaze generation.
- We conduct experiments on FFHQ datasets. The experiment demonstrates our method achieves better gaze-controllable face generation performance than compared methods.

## 2 RELATED WORK

### 2.1 Eye-Control Face Generation

Face generation methods often prioritize controlling semantic information [18, 33], such as appearance and hair color, but frequently overlook gaze control. One closely related task is gaze redirection, where the goal is to generate face images aligned with a specified gaze direction. Deepwarp [15] employs a deep network to learn warping maps between pairs of eye images with different gaze directions. Yu [38] utilizes a pretrained gaze estimator and trains a network to generate eye images. He [10] utilize GAN for gaze redirection and synthesize large-scale gaze data by performing gaze redirection tasks on one eye image. Ruzzi [29] train a NeRF model based on given gaze directions. They generate 3D face models but produce low-quality rendered images. However, these methods share a common limitation in that they all require images with gaze annotations for training, significantly restricting their applicability. This requirement poses challenges for most image datasets where obtaining an annotated gaze may be impractical.

### 2.2 Text-to-Image Generation

Text-to-image generation shows significant achievement within the context of generative adversarial[7]. They leverage text description as a condition and train a conditional GAN[19] based on a pair of image-text samples[26]. Later, stacked structure[39] and attention mechanism[36] are proposed to improve the quality of image generation. Crowson et al. [3] leverage CLIP [24] embeddings to guide image editing use VQ-GAN [6]. They train the model over large-scale data for diverse generations. Recently, there are many large text-to-image models such as Imagen[30], DALL-E2 [25], and Stable Diffusion[27]. These models leverage the diffusion model and produce unprecedented generation.

## 3 TOG: TEXT OF GAZE DATASET

We utilize large language models (LLMs) to generate text-of-gaze (ToG) dataset. Our dataset contains over 90k text descriptions.

**Table 2: We present text descriptions from the ToG dataset. We define two levels of precision and leverage LLMs to generate different descriptions for each precision level. The last two columns present the corresponding gaze and head labels.**

| Precesion | Descriptions | Head label | Gaze label |
|---|---|---|---|
| Low | The person's head turns left, gaze shifts left and slightly up | $(60°, 0°)$ | $(109°, 10°)$ |
| | The person kept the head and the gaze straight ahead | $(0°, 0°)$ | $(0°, 0°)$ |
| | The person tilted the head right, directing the gaze sharply downwards | $(-70°, -70°)$ | $(-115°, -20°)$ |
| High | The person's head turns significantly left, remaining level, while the gaze shifts sharply left and slightly upwards, indicating keen interest or examination. | $(60°, 0°)$ | $(109°, 10°)$ |
| | The person maintained a direct, steady posture, with the head and the gaze fixed straightforward, indicating intense focus. | $(0°, 0°)$ | $(0°, 0°)$ |
| | The person directed the head significantly to the right and downward, while the gaze extended far right, with a minimal decline. | $(-70°, -70°)$ | $(-115°, -20°)$ |

## 3.1 Overview

Text-to-image tasks typically rely on paired text-image data $\{I_i, T_i\}$ for training, where $I_i$ denotes images and $T_i$ represents the corresponding text description. However, the collection of such data is both time-consuming and costly, as annotators are required to manually generate text descriptions for each image.

Recently, LLMs attract significant attention and demonstrate remarkable capabilities in natural language. This inspires us to explore the potential of leveraging LLMs for data generation. In this section, we propose a cost-effective method for generating text data using LLMs. We consider following two questions:

1) *How can LLMs be utilized for the data generation?* While recent LLM models exhibit proficiency in understanding images, their functionality may not be entirely convincing. In our approach, we disentangle gaze information from images. Rather than generating paired data $\{I_i, T_i\}$, we generate paired data $\{g_i, T_i\}$, where $g_i \in \mathbb{R}^2$ represents the gaze in the form of (Yaw, Pitch). Compared to high-dimensional image data, gaze directions offer greater accuracy and ease of interpretation. Additionally, we can further estimate gaze from images to establish the connection between text and images.

2) *How can the validity of the generated data be confirmed?* To assess the quality of the generated data, we sample text descriptions from our dataset and conduct a user study for evaluation.

Next, we introduce the LLM-based data generation method in the section 3.2. We present an overview of the ToG dataset statistics and discuss the user study evaluating our dataset in the section 3.3.

## 3.2 Text Generation using LLMs

We aim to utilize LLMs to generate text descriptions from gaze. However, relying solely on gaze values may not accurately capture human facial features. Human gaze is influenced by both head pose and eye rotation, introducing ambiguity that can result in unclear text descriptions. Therefore, we further decompose the gaze into head pose and eye pose to address this challenge.

**Dense Gaze Sample Generation:** We respectively define the ranges for head pose and eye pose, and calculate gaze based on these values [5]. Specifically, we define head pose in both the yaw and pitch axes, sampling from the range of $-70°$ to $70°$ at intervals of $10°$. This results in $15 \times 15$ head rotation samples. We also sample eyeball rotation from the range of $-50°$ to $50°$ at intervals of $10°$,

generating $11 \times 11$ eye rotation samples. By combining these values, we generate a total of $27,225$ gaze samples. To ensure visibility of the eyes, we exclude gaze samples that fall outside the range of $-70°$ to $70°$ in the pitch axis and $-120°$ to $120°$ in the yaw axis [41]. Consequently, we obtain a total of 23,887 samples.

**Prompt Design:** We further employ LLMs to generate text description from gaze. We design the following prompt:
*"Imagine describing someone's head and gaze movement. Given four numbers representing yaw and pitch for both head and gaze (positive for left/up, negative for right/down), craft a detailed description. Your narrative should:*
*1. Avoid numerical values for angles;*
*2. Offer rough direction descriptions regardless of extent;*
*3. Be one sentence and within 10 words;*
*4. Begin with 'The person' and describe 'the head' and 'the gaze' impersonally."*

The first requirement ensures that the LLM does not produce annotations based on simple templates such as *"The person's gaze is 60 degrees upward"*. The second and the third requirements set limitations to the precision and the length of the annotations. We usually describe gaze behavior roughly or accurately in different scenarios. The two requirements make LLM to generate only low-precision descriptions, such as *"The person tilts the head right, gazing up and left"*. Besides, we design another two requirements which require LLM to generate high-precision annotations:
*"2. Offer precise direction descriptions with a clear extent;*
*3. Be concise, between 20 and 30 words;"*
It allows LLM to generate detailed pose annotation like *"The person gently rotated the head to the right and deeply dipped it, whereas the gaze darted left before sharply descending, aligning slightly off-center"*. In addition, human faces have many attributes such as gender and age. To make the descriptions focused on describing pose, we set limitations to the narrative format with the fourth requirement. Another advantage is that we can easily replace the pronouns in sentences like replacing *"The person"* with *"The boy"*, *"The girl"*, or *"The farmer"*. It expands the capability of our method for diversification.

To further expand the diversity of the dataset, we ask LLM to generate three different descriptions in high precision for each sample simply by adding *"Please give me three different descriptions."*

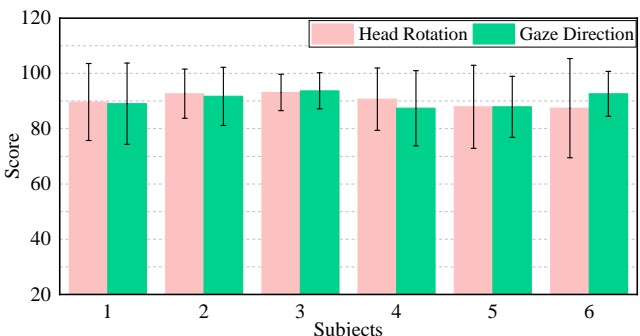

**Figure 2: Visualization of head and gaze scores of different subjects.**

to the prompt. We show some examples of different precisions in Table. 2. The prompt encourages LLM to replace words, especially the ones describing the extent. Please refer to the supplementary material for more examples.

### 3.3 ToG Dataset

We utilize ChatGPT-4 Turbo for data generation. In total, we sample $23,887$ gaze values, considering variations in both eye rotation and head rotation. For each gaze value, we generated four descriptions, one with low precision and the other three with high precision given that low-precision descriptions exhibit limited variability. Consequently, we obtained four descriptions for each sample, resulting in a total of 95,548 annotations. Each sample in our ToG dataset is denoted as $\{\mathbf{g}_i, \mathbf{h}_i, \mathbf{T}_i\}$, where $\mathbf{g}_i$ represents the gaze value, $\mathbf{h}_i$ denotes the head pose, and $\mathbf{T}_i$ is the text description.

We compare our ToG dataset with existing human-centered text-to-video/image datasets in Table. 1. BABEL [22], HumanML3D [8], PoseScipt [4] focus on text pose generation. Text2FaceGAN [20] and CelebAText-HQ [34] are for text face generation. Our ToG dataset has over 90k text descriptions which are comparable with existing datasets. Besides, our ToG dataset mainly focuses on text gaze generation which the existing datasets can not be used for.

To evaluate the generated dataset, we randomly select 20 samples from our ToG dataset. We get the famous gaze dataset ETH-XGaze dataset [41], which provides corresponding gaze values and image pairs. We match each description with an image in ETH-XGaze dataset by finding the closet pose labels, *i.e.*, gaze value and head pose. As the result, we acquire image-text pairs.

We invited six users to rate the correspondence between text descriptions and images on a hundred-point scale, with lower scores indicating poorer correspondence. Specifically, we asked them to score the correspondence between the image and the text description of the head, as well as between the image and the text description of the gaze. We show the head and gaze scores from different users in Fig. 2. The results show that both scores for head and gaze are around 90 with small variances, which indicates that our text descriptions match both head and gaze values well. The good results validate the quality and accuracy of the generated data.

## 4 METHOD

### 4.1 Overview

Given a text description $\mathbf{T}$ describing human gaze behavior, our objective is to generate face images $\mathbf{I}$ that align with the description. In the previous section, we present the ToG dataset, which provides paired data $\{\mathbf{g}_i, \mathbf{h}_i, \mathbf{T}_i\}$. Additionally, we have an image dataset $\{\mathbf{I}_i\}$ containing diverse face images. Matching face images with text descriptions is a challenge in this task. While some gaze estimation datasets offer paired data $\{\mathbf{g}_i, \mathbf{h}_i, \mathbf{I}_i\}$, where the gaze labels are obtained through calibration, these datasets typically lack diverse face appearances.

To tackle this challenge, we propose a two-stage text-to-face image generation method. Our method contains a sketch-conditioned face diffusion module and a model-based sketch diffusion model. We define a face sketch based on facial landmarks and eye segmentation, and the face diffusion model generates face images from face sketch. Importantly, the model is trained without gaze annotation. The sketch diffusion model, on the other hand, learns gaze patterns from text descriptions. To convert the gaze pattern into a face sketch, we utilize a 3D face model and compute the face region information based on eye and head rotation.

### 4.2 Text-to-Gaze Generation

We first build a diffusion model to learn gaze pattern from text discription. In general, we learn a diffusion model $\mathcal{G}$, where $\{\mathbf{g}, \mathbf{h}\} = \mathcal{G}(\mathbf{T})$. We obtain feature embeddings from the text description and propose a text attention module (TAM) to capture head and gaze information in this section.

Specifically, given a text $\mathbf{T}$, we first convert it into feature embedding $\{f_{text}^i, i = 1...L\}$, where $L$ is the length of $\mathbf{T}$. We utilize CLIP [24] to obtain the embedding feature for each word. To effectively extract pose information from the text, we propose the TAM. Our main idea is to extract gaze-related and head-related text information from $\{f_{text}^i\}$. In particular, CLIP preserves feature similarity for two similar words due to contrastive learning. We acquire additional word embedding for the words 'gaze' and 'head' via CLIP, denoted as $f_{gaze}$ and $f_{head}$ respectively. These embeddings are used to aggregate gaze-related and head-related information from $f_{word}^i$. TAM is consisted of two cross-attention modules and one self-attention module. The inputs of the two cross-attention modules are $\{f_{gaze}, f_{text}^i\}$ and $\{f_{head}, f_{text}^i\}$ respectively, allowing them to aggregate text features related to gaze and head. Additionally, we input $\{f_{text}^i\}$ into a self-attention module to extract sentence feature. TAM combines the three extracted features as the output, which are then used as conditions to generate gaze and head via diffusion models.

The diffusion model $\mathcal{G}$ generates $\{\mathbf{g}, \mathbf{h}\}$ from $\mathbf{T}$. We then convert the gaze information into an intermediate representation, face sketch, for face image generation. In detail, we leverage the 3D face model FLAME [17] in our work. We rotate the 3D face model based on $\{\mathbf{g}, \mathbf{h}\}$ and project it into 2D space. We acquire 2D landmarks and connect these landmarks of different region to generate face sketch. Considering the importance of eye region, we highlight the eye region and pupil using different values. We will explain the advantage of the face sketch in the next section.

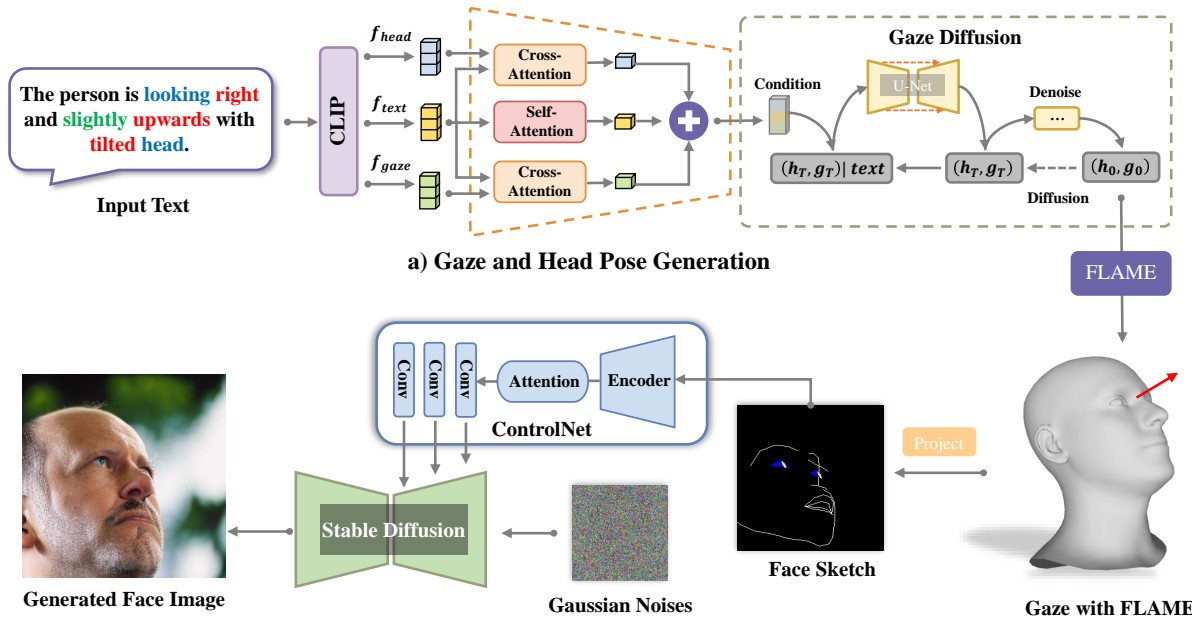

**Figure 3: The model operates in two distinct stages: pose generation and face generation. During the first stage, the text description is initially fed into the CLIP model to obtain word embeddings. These embeddings are then processed through our TAM module as the condition to gaze diffusion module. The gaze diffusion module generates head pose and gaze direction matched with input text. Subsequently, these poses are utilized to rotate a 3D face model into the predicted orientation. The rotated model is then projected into a two-dimensional space, resulting in the creation of a sketch. In the second stage, the face image is meticulously crafted using a diffusion model, which is specifically conditioned on the sketches generated in the previous stage. This two-tiered approach ensures a coherent and detailed synthesis of facial images.**

## 4.3 Gaze-Controllable Face Generation

We aim to generate face image using gaze information $\{\mathbf{g}, \mathbf{h}\}$. Recent gaze redirection methods direct input gaze information as condition and generate satisfied images. However, it means we need paired data $\{\mathbf{g}, \mathbf{h}, \mathbf{I}\}$ for training. In this section, we first convert gaze information into face sketch using 3D face model, and then perform face-sketched condition face generation. This pipeline enables us to perform gaze-controllable face generation without gaze label.

As shown in Fig. 3, the face sketch is made of facial layout landmarks and iris segmentation, which give information on head pose and gaze direction. We use the segmentation map to highlight eye region since the eye region is important for the gaze-controllable generation. More concretly, given an image $\mathbf{I}$, we perform facial landmark detection and eye segmentation on $\mathbf{I}$ [2, 21]. We connect landmarks and conbine it with eye segmentation to produce face sketch. As the result, we obtain the paired data $\{\mathbf{I}, \mathbf{I_s}\}$, where $\mathbf{I_s}$ represents the face sketch. We use the data to train a conditional diffusion model for the gaze-controllable face generation.

We use the ControlNet [40] to generate face images from facial sketches. We learn the marginal distribution of face image $\mathbf{I}$ conditioned on the sketch $\mathbf{I}_{sketch}$.

$$\mathcal{L} = \mathbb{E}_{\mathbf{z}_0, \mathbf{t}, \mathbf{c}_t, \mathbf{c}_f, \epsilon \sim \mathcal{N}(0,1)} \left[ \| \epsilon - \epsilon_\theta \left( \mathbf{z}_t, \mathbf{t}, \mathbf{c}_t, \mathbf{c}_f \right) \|_2^2 \right] \quad (1)$$

We use Eq. (1) as the objective function while we learn added noise $\epsilon$ rather than image $\mathbf{x}_0$. We also fix the Stable Diffusion as the backbone and only train the control module in training [40].

In the inference stage, given gaze information $\{\mathbf{g}, \mathbf{h}\}$, we leverage a 3D face model to generate the face sketch. We rotate the 3D face model based on $\{\mathbf{g}, \mathbf{h}\}$ and project it into 2D space. We acquire 2D face and eye landmarks, then connect them for face sketch. We also highlight the eye region as eye segmentation map based on eye landmarks.

## 4.4 Implementation Details

Our proposed baseline is implemented by Pytorch. We generate $256 \times 256$ image from the text description. We use ControlNet [40] as the diffusion model for face image generation. We use transformer with **6** layers **8** heads for all three branches in the conditioning part of Text-to-Gaze diffusion model. The Text-to-Gaze diffusion model is built based on Latent Diffusion Model [27]. We adopt the Adam optimizer and the learning rate of 1.0e-06 for training the two models. We use batch size of 8 with 20 epochs for face diffusion model and batch size of 16 with 10 epochs for text-to-gaze diffusion model. We use Stable Diffusion v2.1 as the backbone in the face diffusion model. We train the face diffusion model and the text-to-gaze diffusion model for 24 hours and 7 hours separately on one RTX 3090 GPU.

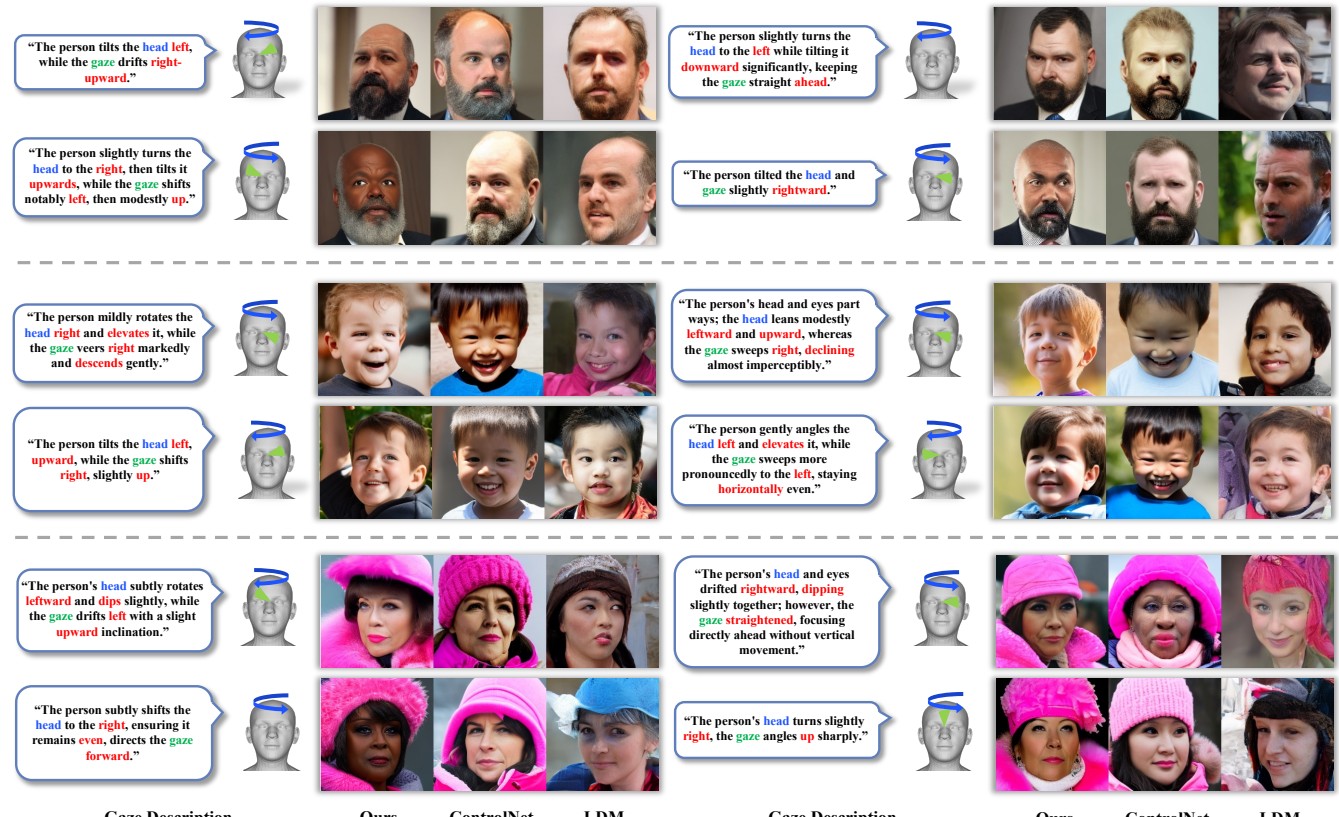

**Figure 4: Visulization of generated images from our model and two baselines [6, 40]. We show 12 sets of comparisons in three styles indicated by different colors. In each set, pose description is listed on the left and the images generated by our model, ControlNet, and LDM are listed side by side on the right. *"head"*, *"gaze"* and directional words are highlighted for better visualization. While images from ControlNet and LDM are meaningful, they often fail to match the head pose, gaze direction, or both specified in the text. Our approach effectively captures and aligns head pose and gaze details with the textual descriptions, maintaining high image quality.**

**Table 3: Comparison to other SOTA methods on the FFHQ dataset in terms of image quality (IS, FID, KID), correspondence of text and image (CLIP-score, User Preference for Head and Gaze). KID\* stands for KIDx1000. Bold indicates the best number.**

| Method | IS ↑ | FID ↓ | KID* ↓ | CLIP-score ↑ | User Preference (Head) ↑ | User Preference (Gaze) ↑ |
|---|---|---|---|---|---|---|
| LDM | 3.89 | **55.45** | 45.34 | 27.36 | 0.07 ± 0.07 | 0.05 ± 0.05 |
| ControlNet | 7.11 | 61.67 | 35.94 | **30.84** | 0.11 ± 0.07 | 0.11 ± 0.10 |
| Ours | **7.22** | 59.91 | **35.73** | 30.63 | **0.76 ± 0.16** | **0.75 ± 0.20** |

## 5 EXPERIMENT

### 5.1 Datasets

**ToG Dataset.** We divide our ToG dataset into a training set and a test set. The training set ratio is 0.9. We use the training set to train the text-to-gaze generation model.

**FFHQ Dataset.** The face generation model undergoes training utilizing the FFHQ dataset [14], a collection of 70,000 superior-quality facial images sourced from the internet, frequently employed in generative model training [14, 28]. FFHQ's native resolution stands at 1024 × 1024, while our experiments employ its images

resized to 256 × 256. We get the style prompts of FFHQ dataset with BLIP [16]. To evaluate our model, we split the FFHQ dataset into a training part and a test part. The training part ratio is 0.9.

**Sketch Generation.** The generated sketch includes face layout landmarks and iris segmentation. We get face landmarks using face-alignment [2]. We get iris landmarks and eyelid landmarks with the eye landmark detector [21]. We connect the landmarks of different parts and make it a sketch. We further mask out the invisible iris region with eyelid landmarks to generate a realistic sketch. We use the sketches generated from a 3D face model in the reference stage. To mitigate the gap between the sketches in

the training stage and the inference stage, we align the sketches by setting the camera parameters in the 3D face model same as FFHQ dataset. Specifically, we first get the camera parameters used in FFHQ dataset by fitting the images to the face model and then fix the parameters when generating new sketches.

## 5.2 Comparison on Text-conditioned Generation

**Baselines.** The existing text-to-image generation methods [6, 25] are not optimized specifically for gaze image generation. As far as we know, we are the first work to do gaze-controllable face generation with natural language. To make a fair comparison, we first match the images in the FFHQ training set with the text descriptions in our ToG dataset and then train a latent diffusion model (LDM) [6] and a ControlNet model [40] using the matched data. Specifically, we apply a pre-trained pose estimator to gauge head poses and gaze directions on FFHQ's training set [44]. Then we find the closest pose labels and the corresponding descriptions to the estimated ones. Finally, We match the selected descriptions with the images to get the training data. For the LDM model, we combine the pose descriptions with the style prompts as the input. The LDM model is trained from scratch. For the ControlNet model, we input the style prompt as [40] and adopt the control network to accept the pose descriptions as the additional input condition. The ControlNet model also uses Stable Diffusion v2.1 as the backbone. Both models are trained with a batch size of 8 in an end-to-end way for 30 epochs.

**Metrics.** We report the standard evaluation metrics Inception Score (IS) [31], Fréchet Inception Distance (FID) and Kernel Inception Distance (KID) [11] on the FFHQ test set to evaluate the quality of generated images. We report KID* which is 1000 x KID to make a more precise comparison. We also report CLIP scores [23] on the same data to evaluate the correlation between prompts and generated images. We use DDIM as the sampler and use 50 steps to sample each image. We calculate FID and KID metrics on FFHQ test set. We combine style prompt and pose description as one prompt when calculating CLIP scores. We further perform a user study to evaluate our model's effectiveness. We randomly chose 20 generated images with the same pose prompts for our model, ControlNet, and LDM. We get 20 sets with a total of 60 images. Twelve independent blinded users were invited to join the study and they were shown 20 side-by-side images each generated by our model, ControlNet, and LDM. The users were given the task of selecting the image which matches the given descriptions best in terms of gaze direction and head pose separately. If none of the images matches the description, the user could select a "None" option. We report user preferences for head and gaze according to the user study. Note that CLIP model is not optimized specifically for precise pose descriptions. Therefore, CLIP score does not reveal how well the generated images match the pose descriptions. We deem that head fidelity and gaze fidelity from the user study are more suitable metrics for our work.

**Qualitative Results.** The qualitative results are shown in Fig. 4. We give 12 sets of images and their pose descriptions. Thanks to the capability of ControlNet, our model can also control the style of the generated images. We split the samples into three style collections

for better visualization. The pose descriptions and the images are listed side by side. All three models can generate meaningful face images. The images generated by our model and ControlNet are of good quality while the results of LDM have some artifacts. It is because our model and ControlNet only trained an additional network to add additional conditions leaving the Stable Diffusion backbone unchanged. Therefore, they always generate images with a similar quality to Stable Diffusion. However, the whole LDM model is optimized in the training, which might make the model easily overfit the training data and unstable. Even though the generated images from ControlNet and LDM are meaningful, they do not follow the head pose, gaze direction, or both depicted in the input text. Both baseline models can generate various head poses and gaze while the contents are not related to the text descriptions. Their disability to render correct gaze images under text instructions shows that capturing geometric information from text descriptions directly is hard. To solve the problem, we design two stages including pose generation and face generation. The pose generation model estimates the head pose and gaze direction from text and transfers the pose information to face sketches explicitly through a 3D face model. The face generation model takes the face sketches as input and renders face images. The two-stage design alleviates the difficulty of direct matching between text and images. As shown in the results, our generated images effectively capture the head and gaze information and maintain good quality simultaneously.

**Quantitative Results.** We show the quantitative results in Table 3. Our model achieves the best performance on Inception Score and KID and comparable performance on FID compared with the other two baselines. It indicates that our model can generate high-quality face images with the additional eye-controllable function. Our model achieves a comparable result on CLIP score with ControlNet while outperforming LDM by a large margin. We show the user preferences for head and gaze in the last two columns. The results show that users prefer our generation most of the time and our generated images match the pose descriptions well.

The good performance is attributed to our newly designed two-step image generation. Our high-performance two-step image generation process starts with obtaining predicted head pose and gaze directions using our text-to-pose model, trained on the ToG dataset. This model effectively extracts gaze and head pose information from text, generating them within a reasonable range. We then use the FLAME model to simulate head and eyeball rotations, producing a rotated 3D face model for sketch generation. In the second stage, we use a diffusion model trained with ControlNet to generate images that accurately incorporate the predicted head pose and gaze data. This ensures that the head and gaze details are consistently preserved from the initial text description through to the final images.

## 5.3 Ablation Study

**Can we replace sketch with pose labels?** We advocate for the use of face sketches in facial generation, owing to the scarcity of gaze labels. Numerous studies have focused on head and gaze pose estimation. This prompts the question: could these estimated poses replace sketches? To explore this, we applied a pre-trained VGG [32]-based gaze estimator [44] to gauge head poses and gaze

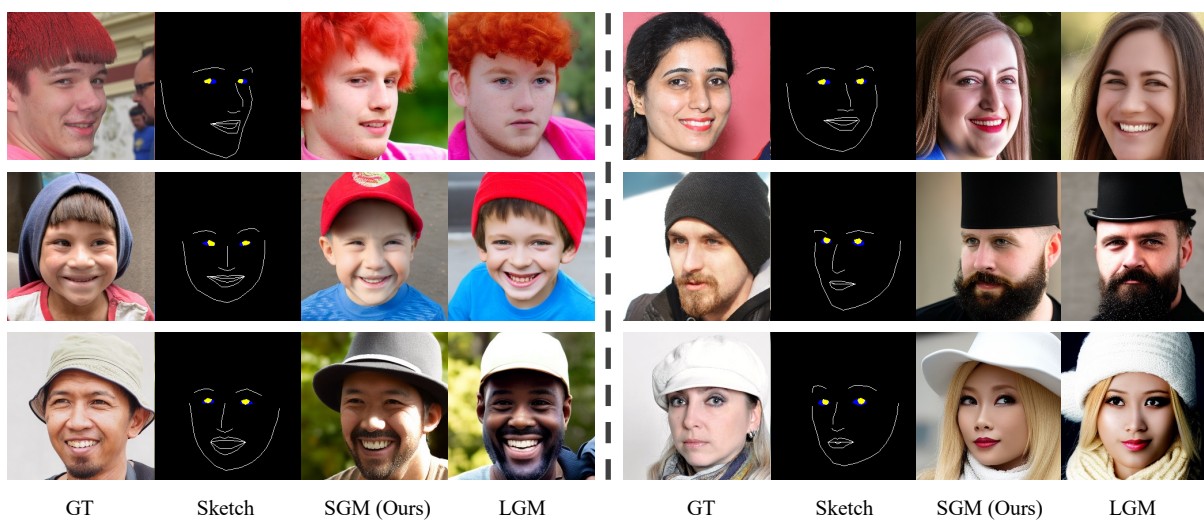

|  |  |  |  |  |  |  |  |
| GT | Sketch | SGM (Ours) | LGM | GT | Sketch | SGM (Ours) | LGM |

**Figure 5: Visualization of generated images from Label-Guided Model and Sketch-Guided Model. The Label-Guided Model (LGM) falls short of accurately reconstructing true geometric information. On the other hand, the Sketch-Guided Model (SGM) effectively maintains geometric integrity throughout the generation process.**

**Table 4: Comparison of LGM and SGM in terms of head error and gaze error in degree on the test set of FFHQ dataset.**

| Method | Head ↓ | Gaze ↓ |
|---|---|---|
| LGM | 4.37 | 4.79 |
| SGM (ours) | **3.62** | **4.35** |

directions on FFHQ's training subset. We adapted the condition encoding segment of a ControlNet model to accept pose inputs, keeping other components intact. Subsequently, we developed a label-guided model. We refer the label-guided model and our sketch-guided model to LGM and SGM separately. LGM is trained on FFHQ training set with estimated pose labels. Similar to our face generation model, it is trained with a batch size of 8 for 20 epochs. We evaluate the models on the FFHQ test set.

We show the qualitative comparison with our sketch-guided model in Fig. 5. The label-guided model shows limited control of head pose and gaze direction in the generation. We estimate the head poses and gaze directions of the generated images from the two models using another pre-trained ResNet50 [9]-based gaze estimator from [44]. We calculate the head error and gaze error in degree and report them in Table. 4. The results show that our sketch-guided model generates images with higher accuracy in head pose and gaze direction. These results underscore the efficacy of our approach, confirming the validity of our choice to rely on sketch-guided modeling.

**Effect of Text Attention Module.** To show the effect of our designed text attention module, we train a pose generation model with only a self-attention module under the same training settings for comparison. We further test two different ways (adding and concatenation) to fuse the features from different branches. We report the generated angle errors on the test set of our ToG dataset.

**Table 5: Ablation on TAM module in terms of generated head error and generated gaze error in degree on the test set of ToG dataset.**

| Method | Generated Head ↓ | Generated Gaze ↓ |
|---|---|---|
| w/o TAM | 16.94 | 21.62 |
| w/ TAM (concat) | 15.45 | 20.28 |
| w/ TAM (add) | **15.23** | **20.00** |

The angle errors are calculated via cosine similarity between the estimated poses and ground truth poses. We show the results in Table. 5 The two models with TAM get comparable results and achieve better performance compared with the pure self-attention model. The results indicate that our TAM is more effective in capturing pose information from text descriptions.

## 6 CONCLUSION

In our study, we introduce an innovative task of gaze-controllable face generation, driven by textual descriptions of human gaze behaviors. We present the pioneering text-to-gaze dataset, featuring over 90k descriptions that align with various gaze behaviors. This dataset encompasses a wide range of gaze directions, enriched by diverse natural language annotations derived from Large Language Models (LLMs). Building upon this dataset, we develop a novel two-stage face diffusion model. Initially, we create detailed face sketches, which are then transformed into face images using conditional diffusion models. Additionally, our approach infers face sketches from text descriptions, incorporating a 3D face model prior. Significantly, our method operates without the need for explicit gaze annotations and can generate gaze-controllable face images effectively. The efficacy of our approach is further demonstrated through experiments conducted on the FFHQ dataset.

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
