# OpenReview forum: "TextGaze: Gaze-Controllable Face Generation with Natural Language"
_acmmm.org/ACMMM/2024/Conference — MM2024 Poster_

### Official Review · Reviewer_n13P · 2024-05-08

**Rating:** 4
**Confidence:** 3

**Summary:**

This paper aims to generate faces of varying gazes and head poses by controlling the input texts. The proposed method is composed of two diffusion model, a text-to-gaze diffusion model (to convert text inputs into gaze representations) and a sketch-to-image diffusion model (to convert gaze representations into realistic faces). To supervise the text-to-gaze diffusion model, a ToG (Text of Gaze) dataset is first collected by converting gazes of numerical values into text descriptions using the LLM. The text-gaze pairs in ToG are then utilized to train the text-to-gaze diffusion model. To achieve sketch-to-image translation, this paper adopts the ControlNet conditioned on the sketch.

**Strengths:**

① The paper is well organized and easy to understand.

② The text-based editing provides a flexible user interface which enjoys a wide range of application. The pipeline proposed in this paper can be easily adapted to other human face attributes (using VLM instead of LLM), such as the age, gender, skin color and face shape.

**Limitations:**

### Weakness and Question

① Could you please give more examples on the ToG (Text of Gaze) dataset? Do neighboring labels of head and gaze have distinct descriptions? For instance, how do the descriptions differ between samples with a head label of $(50^{\circ},50^{\circ})$ and those with a head label $(50^{\circ},60^{\circ})$.

② What are the representations of the head pose $h$ and the gaze $g$? Are they implicit features or 3DMM coefficients? Why is the features of relatively low dimension diffused in the latent space?

③ Could you please provide a more detailed explanation on how the 3D face model is projected into the 2D sketch, and how the sketches from the 3D model and the FFHQ images are aligned? Are there global camera parameters available in the FFHQ dataset?

④ Some minor comments:

+ The texts in Line 144-152 and Line 153-161 are essentially the same.

+ Line 499 image $\rightarrow$ images

+ Potential Limitation: the proposed methods can not achive fine-grained gaze control considering the ambiguity of the text inputs.

**Suitability:**

3

---

### Official Review · Reviewer_4Jqo · 2024-05-23

**Rating:** 2
**Confidence:** 4

**Summary:**

This paper presents a gaze-controllable face generation from textual descriptions. Key contributions include: introducing the first text-of-gaze (ToG) dataset with over 90k descriptions by using Large Language Models (LLMs) and developing a two-stage text-to-face generation method that leverages 3D face models and a text attention module. Experimental results demonstrate that the proposed method outperforms existing methods in generating gaze-controllable face images.

**Strengths:**

1. **Detailed introduction of ToG Dataset**: ToG is the first well-designed text-to-gaze generation dataset using LLMs support, which opens up new possibilities for dataset annotation. The dataset includes both head pose and eye gaze information, also considerably designed low-precision and high-precision text descriptions. Looking forward to the dataset being released.
2. **Rich experiments**: This paper includes several experimental results and evaluations to invalidate the effectiveness of the proposed method.

**Limitations:**

1. **Novelty**: The main contribution is the ToG dataset, which contains paired data: text descriptions, gaze value, and head pose. For the two-stage text-to-face image generation model, the text-to-gaze model adopts CLIP, cross-attention module, self-attention module, and FLAME to generate face sketch; the sketch-to-face model relies heavily on existing methods: ControlNet and Stable Diffusion.

2. **Clarity**: There are several grammar mistakes and typos.

   [Table 2] "Precesion"->"Precision";  [Line 510] "concretly"->"concretely"; [Line 512] "conbine "->"combine "; [Figure 4] "Visulization"->"Visualization"

3. **Missing details**:

   Sections 4.2 and 4.3 are unclear and lack sufficient detail. The last paragraph in Section 4.2 is almost the same as the last paragraph in Section 4.3 and does not focus on text-to-gaze generation, which would be better to combine into Section 4.3.

   - [Line 458-460] How to project the 3D face into the 2D space and get the face sketch?
   - [Line 508-510] How to get facial layout landmarks and iris segmentation maps?
   - [Line 510] Where is the "given image **I**"? It would be better to show the face image "**I**" in Figure 3.

4. **Unclear statements**: Certain phrases are unclear without detailed elaboration, which makes this paper difficult to understand. For instance:

   - [Line 684] What does the phrase “style prompts” mean?

5. **References**: The reference format is not consistent. [23] and [24], [27] and [28] are the same paper.

6. **Limitation discussion**: Limited discussion on limitations and future work. It would be better to provide a more explicit discussion of the limitations of the proposed method. Acknowledging potential shortcomings and future works can provide a broader perspective for readers.

**Suitability:**

3

---

### Official Review · Reviewer_drv8 · 2024-05-25

**Rating:** 4
**Confidence:** 3

**Summary:**

This paper clarifies the concept of a novel kind of generation subtask: text-driven gaze-controllable face generation, for the development of user-friendly custom facial image generation. It provides the first text-to-gaze dataset containing more than 90k text descriptions via ChatGPT-4 Turbo. This paper also proposes a gaze-controllable text-to-face method, which includes a sketch-conditioned face diffusion module and a model-based sketch diffusion module. The face diffusion module generates face images from the face sketch, and the sketch diffusion module employs a 3D face model to generate face sketch from text description. Experiments on the FFHQ dataset show the effectiveness of this two-stage generative method.

**Strengths:**

This paper clarifies the concept of a novel kind of generation subtask: text-driven gaze-controllable face generation, for the development of user-friendly custom facial image generation. It provides the first text-to-gaze dataset containing more than 90k text descriptions via ChatGPT-4 Turbo. This paper also proposes a gaze-controllable text-to-face method, which includes a sketch-conditioned face diffusion module and a model-based sketch diffusion module. The face diffusion module generates face images from the face sketch, and the sketch diffusion module employs a 3D face model to generate face sketch from text description. Experiments on the FFHQ dataset show the effectiveness of this two-stage generative method.

**Limitations:**

1. This paper simplifies the task of text-driven gaze-controllable face generation and avoids some issues that might limit the diversity of the generated results. As a pioneer in this newly proposed subtask, these simplifications are understandable, but they indeed limit the diversity of the generation. For example, the face sketch is generated by projection from a 3D face model, while head labels and gaze labels are both in the same form of (Yaw, Pitch), which only involves two rotational degrees of freedom in the three-dimensional space. What about ROLL?

2. The generation method in this work appears not to involve identity information, which is extremely important for facial image synthesis, especially given that one of your motivations is to provide a more user-friendly controllable generation method. Thus, the prompts required are limited to “The person …”.
“To make the descriptions focused on describing pose, we set limitations to the narrative format with the fourth requirement.” This seems to provide an explanation, but it does not address the issue of uncontrollable identity, because there is no guarantee that your work in this paper can still achieve good generation results when incorporating identity information without experiments.
”Another advantage is that we can easily replace the pronouns in sentences like replacing ‘The person’ with ‘The boy’, ‘The girl’, or ‘The farmer’. It expands the capability of our method for diversification.” You can, but you haven’t, at least not for your generation method. The examples presented in Figure 4 do not include these pronouns.

3. The existing experimental section appears to be somewhat insufficient in scope. In Section 5.2, this paper only selects LDM and ControlNet as "SOTA" models. Are there no other text-driven facial synthesis methods based on LDMs/ControlNet that are more focused on the facial generation task? The experimental section would be more convincing if additional metrics regarding “non-gaze-controllable” general text-driven facial generation models could be provided.


4. There is also room for improvement in the writing, for example:
- In Section 2.1, please review carefully the description of previous works, e.g., “He [10] utilize …” and “Ruzzi [29] train …”. You may need to add “et al.”
- In Section 5.1, when listing the TOG and FFHQ datasets together, it would be more aesthetically pleasing to use the same indentation for both, rather than the default formatting.
- In Section 5.1 Sketch Generation, it seems somewhat amateurish to describe in this way: “We get…We get…We connect…We further mask…We use…”. This kind of issue is present throughout almost the entire 3. TOG and 4. Method sections, so please carefully review your article and improve your wording in the final draft to align with the standards of academic English.
-The blank space below Figure 2 appears to be excessive.
-In Section 5.3 Can we replace…, please do NOT use the past tense when introducing your own work.

**Suitability:**

3

---

### Meta-Review · Area_Chair_44G8 · 2024-07-02

**Recommendation:** Accept (Poster)
**Confidence:** 5

**Metareview:**

The paper receives three borderline-accept recommendations from the three reviewers. They acknowledged the novel design and promising experimental results. The presentation is also very clear. AC agrees with the comments of the reviewers and recommends accepting the paper.